# Replication Variance of African and Asian Lineage Zika Virus Strains in Different Cell Lines, Mosquitoes and Mice

**DOI:** 10.3390/microorganisms9061250

**Published:** 2021-06-09

**Authors:** Tey Putita Ou, Heidi Auerswald, Saraden In, Borin Peng, Senglong Pang, Sébastien Boyer, Rithy Choeung, Myrielle Dupont-Rouzeyrol, Philippe Dussart, Veasna Duong

**Affiliations:** 1Virology Unit, Institut Pasteur du Cambodge, Institut Pasteur International Network, Phnom Penh 12156, Cambodia; oteyputita@pasteur-kh.org (T.P.O.); hauerswald@pasteur-kh.org (H.A.); insaraden@pasteur-kh.org (S.I.); borinpeng@gmail.com (B.P.); psenglong@pasteur-kh.org (S.P.); rithychoeung88@yahoo.com (R.C.); pdussart@pasteur.mg (P.D.); 2Medical Entomology Unit, Institut Pasteur du Cambodge, Institut Pasteur International Network, Phnom Penh 12156, Cambodia; sboyer@pasteur-kh.org; 3URE Dengue and Arboviruses, Institut Pasteur in New Caledonia, Institut Pasteur International Network, Nouméa 98800, New Caledonia; mdupont@pasteur.nc

**Keywords:** Zika virus, African lineage, Asian lineage, vector competence, neonatal mouse infection

## Abstract

Since the epidemic in 2007, studies on vector competence for Zika virus (ZIKV) have intensified, showing that the transmission efficiency varies depending on the vector population, ZIKV strain, and dose of the infectious blood meal. In this study, we aimed to investigate the replication of African and Asian ZIKV strains in vitro and in vivo in order to reveal their phenotypic differences. In addition, we investigated the vector competence of Cambodian *Aedes aegypti* (*Ae. aegypti*) mosquitoes (urban and rural) for these ZIKV strains. We observed a significantly higher pathogenicity of the African ZIKV strain in vitro (in mosquito and mammalian cells), and in vivo in both *Ae. aegypti* and mice. Both mosquito populations were competent to transmit ZIKV as early as 7 days p.i., depending on the population and the ZIKV strain. *Ae. aegypti* from rural habitats showed significant higher transmission and survival rates than those from urban. We observed the highest transmission efficiency for the African ZIKV isolate (93.3% 14 days p.i.) and for the Cambodian ZIKV isolate (80% 14 days p.i.). Overall, our results highlight the phenotypic differences of the ZIKV lineages and the potential risk of ZIKV transmission by *Ae. aegypti* mosquitoes. Further investigations of Cambodian mosquito species and ZIKV specific surveillance in humans is necessary in order to improve the local risk assessment.

## 1. Introduction

Zika virus (ZIKV) is a flavivirus mainly transmitted by *Aedes* mosquito species, representing an example of the emergence of a new arboviral disease. It was first isolated in Uganda in 1947 [1], and for 60 years, infections with this virus were rarely reported. In Asia, ZIKV was first isolated in 1966 in Malaysia from *Aedes (Ae.) aegypti* mosquitoes [2], and the first human infections were reported in 1977 in Central Java, Indonesia [3]. Serological evidence for ZIKV transmission in Southeast Asia was found in surveys in the early 1950s in the Philippines, Malaysia, Thailand, and Vietnam [4]. The first major outbreak was documented in 2007 on Yap island (Micronesia), where over 70% of the population was infected [5]. In 2010, ZIKV was isolated in Cambodia from a single sample through a fever surveillance [6], but no further infections were reported at that time. In 2013–2014, a large ZIKV outbreak occurred in French Polynesia [5]. The virus spread further over the Pacific islands and in 2015, a massive epidemic of ZIKV hit the Americas and became an international public health emergency. For the first time, this outbreak included reports of increased numbers of congenital microcephaly associated with maternal ZIKV infections [7].

Epidemiological and phylogenetic studies discovered that ZIKV strains were grouped by geographic origin, namely, the initial African lineage and the diverged Asian lineage that spread in the 1950s through Southeast Asia [8]. Within this Asian lineage, isolates from the recent ZIKV outbreaks in the Americas clustered together in an American subgroup, whereas the older Asian strains formed a separate subgroup (Appendix A) [9].

Because of the significant public health risk, studies on vector competence for ZIKV intensified globally over the last years [10]. The determined transmission efficiency varies depending on the vector population, the ZIKV strain, and the viral titer of the infectious blood meal. Therefore, these data are difficult to extrapolate, and investigations with local vector populations are necessary for more targeted risk assessment and modelling [11,12,13]. Besides the in vivo experiments in mosquitoes, mammals have been utilized to further understand the biology and pathogenicity of ZIKV [14]. Murine models are valuable tools, and have been utilized to demonstrate the neurotropism and pathogenicity of ZIKV [15].

In Cambodia, the detection of ZIKV relies on incidental findings, because a ZIKV surveillance program for human or mosquito infections is missing. A retrospective prevalence analysis in samples collected from DENV suspected cases from the years 2007–2016 were tested for the presence of ZIKV by RT-PCR [16]. This revealed a low prevalence of 0.2% (*n* = 5) among the 2400 samples tested. The main ZIKV vector, *Ae. aegypti*, is widely distributed in Cambodia [16]. Moreover, Cambodia also has a long history of simultaneous circulation of all four DENV serotypes, with outbreaks occurring frequently every 3–4 years [17]. However, the Dengue National Surveillance System is not suitable to monitor ZIKV infections, as the majority of these infections are asymptomatic and the DENV surveillance focusses on clinical febrile illness in hospitalized children. Thus, the evaluation of the vector competence of Cambodian mosquitoes for ZIKV is important for the risk assessment of the re-emergence of ZIKV in Cambodia.

The aim of our study was the investigation of three ZIKV strains from the two lineages—one belonging to the African lineage and two to the Asian lineage—for their replication in different in vitro and in vivo models. We analyzed their growth in different mosquito cells and in a mammalian cell line. In vivo, we determined the vector competence of urban and rural Cambodian field-collected *Ae. aegypti* populations for the different ZIKV strains and compared the virulence and pathogenesis of the three ZIKV strains in a murine model.

## 2. Materials and Methods

### 2.1. Cells

The simian Vero cell line (ATCC CCL-81) was maintained in Dulbecco’s modified Eagle medium (DMEM; Sigma-Aldrich, St. Louis, MO, USA) supplemented with 10% fetal bovine serum (FBS; Gibco, Waltham MA, USA) and antibiotics (100 units/mL of penicillin and 100 µg/mL of streptomycin; Gibco) at 37 °C under a 5% CO_2_ humidified atmosphere. The three mosquito cell lines used, Aag2 (*Ae. Aegypti*, provided by Julien Pompon, Duke-NUS, Singapore), AP-61 (*Ae. Pseudoscutellaris*, Centre National de Référence des arbovirus, Institut Pasteur, Paris, France), and C6/36 (*Ae. Albopictus,* ATCC CRL-166), were maintained in Leibovitz’s L-15 medium (Sigma-Aldrich) supplemented with 10% FBS, antibiotics, 2 mM L-glutamine (Gibco), and 10% tryptose phosphate (Gibco) at 28 °C under a humidified atmosphere.

### 2.2. Viruses

The African lineage ZIKV strain HD78788 (GenBank: KF383039), originally isolated from a patient in Senegal in 1991 [18], was produced in AP-61 cells. This virus strain is highly adapted to cultivation in various cell lines and the passage history is unknown. The Asian lineage ZIKV strains NC-2014-5132 (GenBank: SRR5309452) and FSS13025 (GenBank: KU955593) were grown in C6/36 cells. Strain NC-2014-5132 was isolated from a patient in April 2014 in New Caledonia, and passaged five times in Vero cells and eight times in C6/36 cells before its use in this study. Strain FSS13025 was isolated on Vero cells from a febrile patient in Cambodia in 2010 [6] and passaged six times in C6/36 cells before use in this study. Virus culture supernatants were stored at −80 °C before titration via FFA and their further use in infection experiments of cells, mosquitoes, and mice.

### 2.3. Mosquito Infection

Mosquito populations from urban Phnom Penh (urban-PP) and from rural Kampong Cham (rural-KC) were collected at the end of rainy season (December 2017) as larvae or pupae, using black plastic cups filled with water and ovitraps (Green Ornament, Osaka, Japan). Field-collected eggs and larvae were reared and fed with half a teaspoon of grounded fish food daily until the emergence of adults (F0 generation). Adult mosquitoes were kept in cages at a temperature of 28 ± 1 °C with 12 h:12 h light:dark cycle, 75 ± 2.5% relative humidity, and with a 10% sucrose solution. Additionally, the females were provided with lab-reared mice for blood meal for 1 h once a week. Adult mosquitoes of the F0 generation were identified morphologically according to the identification key for mosquitoes in Southeast Asia countries [19,20,21]. Only clearly identified *Ae. aegypti* were used for rearing of the F1 generation. Mosquito eggs were collected and hatched in dechlorinated tap water. Emerged larvae were divided into groups of 200 larvae per 1 L water, and were fed daily with 1–2 pellets of rabbit food (Kaytee, Chilton, WI, USA). Adult female mosquitoes of the F1 generation were maintained solely on a 10% sucrose solution before infectious blood meal feeding.

All F1 females aged between 3–5 days were starved by removing the sucrose 24 h before artificial blood meal. For the infection, mosquitoes were sorted into cups containing 30 mosquitoes per cup, and were transferred into the BSL3 laboratory. Artificial blood meals contained 1.4 mL of washed human erythrocytes, viral suspension (final concentration: ~10^6^ FFU/mL), and 5 mM adenosine triphosphate (ATP, Sigma-Aldrich). The ZIKV concentration of the blood-meal was determined by titration on the same day via FFA. For each mosquito population, 300–360 female mosquitos were used for each ZIKV strain. After 30 min of feeding on a pig intestinal membrane attached to feeders of a Hemotek blood-feeding system (Hemotek, UK), the blood-fed mosquitoes were separated, transferred into new cups, and were maintained at 28 ± 1 °C, with a 12 h:12 h light:dark cycle and 80% humidity, and were fed daily with 10% sucrose.

On days 7, 10, 14, and 21 after infection, 30 mosquitoes per virus strain and mosquito population were cold-anesthetized. Salivation was forced by inserting the proboscis for 30 min into a 20 μL tip containing 5 μL of FBS. Afterwards, the saliva-containing FBS was mixed with 45 μL DMEM supplemented with antibiotics and 2.5 µg/mL Amphotericin B (Gibco). The head, legs, and wings of each individual mosquito were stored together in 400 µL phosphate buffered saline (PBS) supplemented with 10% FBS, antibiotics, and Amphotericin B and ceramic beads until homogenization. The remaining body (abdomen and thorax) of each female was processed in the same way. The homogenization was performed using a MagNA Lyser (Roche, Basel, Switzerland) at 6500 rpm for 50 s.

### 2.4. Mice

As a result of availability, the Swiss mice (male and female) used in our study were maintained under pathogen-free and hygiene conditions in the Institute Pasteur in Cambodia in BSL2 facility. One day-old Swiss mice (*n* = 18–20 per group) were infected intracerebrally with ZIKV in order to compare the pathogenicity of different ZIKV strains on mice. The inoculum contained 50 µL of ZIKV suspension (10^5^ FFU/mL) or PBS as the control. The weight and clinical symptoms were monitored daily over 21 days. The level of clinical symptoms was scored as follows: (0) healthy, (1) fever, (2) weaker and more emaciated, (3) limb weakness and back bending, (4) hind-limb or fore-limb paralysis and tremors, and (5) death. Mice infected with NC-2014-5132 and FSS13025 were euthanized at day 7, between day 8 and 10, and at day 21 p.i. with the cervical dislocation method [22], in accordance to the date of mortality observed in mice infected by HD78788 virus. After death or euthanasia, the blood samples and organs tissues were collected. Blood was obtained via cotton swabs and was stored in 400 µL PBS. The organs (brain, heart, intestines, liver, kidney, lung, pancreas, thorax, and primary sex organs) were stored individually in 400 µL PBS, and homogenized with MagNA Lyser at 6500 rpm for 50 s before use.

### 2.5. Real-Time RT-qPCR

The amount of ZIKV titer in the mosquito and mouse organ homogenates was determined by real-time RT-qPCR [23] with RNA extracted using the QIAamp Viral RNA Mini kit (Qiagen, Hilden, Germany).

### 2.6. Focus Forming Assay

The titer of infectious virus was determined by focus forming assay (FFA) and was expressed as focus forming units per milliliter (FFU/mL), as described previously [23]. Titrations were always performed in duplicate. Infected cells visible as foci were automatically counted using an AID ELISpot Reader (Autoimmun Diagnostika GmbH, Strassberg, Germany).

### 2.7. Cell Infection

For the ZIKV growth kinetics, the respective cells (Aag2, C6/36, and Vero) were seeded in 48-well plates with a density of 5 × 10^4^ cells/well. The following day, the ZIKV infection was done with different multiplicities of infection (MOI: 0.01, 0.1, and 1) for 1h at the growth conditions of the respective cell lines. ZIKV stocks used for inoculation were back-titrated by FFA on the same days to ensure correct MOIs. Afterwards, the virus inoculum was replaced by supplemented medium of the respective cell line, as described above with a decreased FBS content (5%). Every 24 h, samples were taken in duplicate and were freshly titrated by FFA. CPE was observed using brightfield microscopy, but was not further quantified.

### 2.8. Ethics Statement

During this study, we followed the World Animal Health Organization (OIE) guiding principles on animal welfare included in the OIE Terrestrial Animal Health Code [24]. Mouse experiment were approved by the National Animal Health and Production Research Institute (NAHPRI) from General Directorate for Animal Health and Production (GDAHP), Ministry of Agriculture, Forestry, and Fisheries. Written consent was obtained from the blood donor volunteers before sampling.

### 2.9. Statistical Analysis

All of the statistical analyses were performed using GraphPad Prism for Windows, version 7.0.1 (GraphPad Software, Inc., La Jolla, CA, USA), and a level for significance of α < 0.05. The statistical test used for each experiment is indicated in the respective results section or figure descriptions.

## 3. Results

### 3.1. ZIKV Growth Kinetic in Mammalian and Mosquito Cell Lines

We analyzed the replication of the three ZIKV strains in the Vero cell line (Figure 1A–C) and the mosquito cell lines C6/36 (*Ae. albopictus*; Figure 1D–F) and Aag2 (*Ae. aegypti*; Figure 1G–I). The mosquito cell lines that were used were *Ae. aegypti* and *Ae. Albopictus,* which are both relevant vectors in Cambodia. The *Ae. pseudoscutellaris* cell line AP-61 was not used for the ZIKV growth curves, as this mosquito species is spread over the South Pacific islands [25] and is therefore not a relevant vector for the Greater Mekong region. The simian cell line was used for comparison, as flavivirus replication has been well studied in these cells [26,27]. The two Asian lineage viruses replicated similarly in all cell lines at all different MOIs. The African strain HD78788 exhibited a cytopathic effect (CPE) from day 2 p.i. in the mosquito cells (red squares in Figure 1D–F for C6/36, and Figure 1G–H for Aag2) and from day 5 p.i. in the Vero cells. Overall, the three virus strains grew similarly in the simian Vero cells. HD78788 grew to the highest titer in Vero cells inoculated with MOI 0.01 (Figure 1A; peak titer 3.38 × 10^6^ ffu/mL on day 4 p.i.) compared with NC-2014-5132 (6.67 × 10^5^ ffu/mL, *p* = 0.018; two-way ANOVA analysis with Tukey’s correction) and FSS13025 (3.93 × 10^5^ ffu/mL *p* = 0.0097). Despite the CPE only seen for the cells infected with HD787888, we observed differences in the max one log in the viral titers of the different ZIKV strains grown in the *Ae. albopictus* cell line C6/36. Compared with strain FSS13025, we observed significantly lower viral titers of HD78788 at MOI 0.01 on day 7 p.i. (*p* = 0.0072; Figure 1D) and at MOI 0.1 on day 4 p.i. (*p* = 0.0357; Figure 1E), day 6 p.i. (*p* = 0.0219), and day 7 p.i. (*p* = 0.0183). However, HD78788 grew to the higher titer in C6/36 cells inoculated with a high virus load of MOI 1 (Figure 1F; peak titer 1.92 × 10^7^ ffu/mL on day 4 p.i.) compared with NC-2014-5132 (4.18 × 10^6^ ffu/mL, *p* = 0.0063) and FSS13025 (1.00 × 10^7^ ffu/mL *p* = 0.0938). In contrast, for the growth differences in the *Ae. aegypti* cells *(*Aag2) infected with HD78788, the CPE was so strong that the virus replication was strongly inhibited and was therefore significant lower at several time points under different MOIs. The African strain HD78788 grew significantly slower, e.g., demonstrated at MOI 1 4 days p.i. (titer 8.13 × 10^5^ ffu/mL; compared with NC-2014-5132 titer 9.80 × 10^6^ ffu/mL, *p* = 0.0251; compared with FSS13025 titer 1.15 × 10^7^ ffu/mL, *p* = 0.0085). Overall, all three ZIKV strains were replicated similarly in simian cells. We observed CPE for only the African ZIKV strain, and the differences in growth were especially prominent in the *Ae. aegypti* cells.

### 3.2. Vector Competence of Cambodian Ae. aegypti Mosquitoes

As we observed significantly higher growth rates for the Asian lineage ZIKV strains in *Ae. aegypti* cells when we compared the vector competence of Cambodian *Ae. aegypti* mosquitoes to these ZIKV strains. We conducted the investigation on two different mosquito populations, one from urban Phnom Penh (PP) and one from rural Kampong Cham (KC). For all of these experiments, ZIKV was detected and quantified using real-time RT-qPCR, and the presence of live ZIKV in saliva was confirmed by FFA.

#### 3.2.1. ZIKV Infection Rate

The infection rates, defined as ZIKV-positive bodies among all blood-fed mosquitoes, were similar (average ≥ 93%) at all-time points (day 7 p.i., 10 p.i., 14 p.i., and 21 p.i.) for all ZIKV strains (Figure 2A,B) for both urban-PP and rural-KC mosquito populations (Appendix A).

#### 3.2.2. ZIKV Dissemination Rate

The dissemination rate of ZIKV HD78788 in urban-PP mosquitoes was significantly higher 7 and 10 days p.i. (100%, Figure 2C) than that of NC-2014-5132 (day 7 p.i.: 80.0%, *p* = 0.0237; day 10 p.i.: 82.8%, *p* = 0.0237) and FSS13025 (day 7 p.i.: 26.7%, *p* < 0.0001; day 10 p.i.: 55.2%, *p* < 0.0001). The dissemination rates were also different between the Asian ZIKVs. Urban-PP mosquitoes infected with FSS13025 had significantly lower dissemination rates than mosquitoes infected with NC-2014-5132 at day 7 p.i. (*p* < 0.0001) and at day 10 p.i. (*p* = 0.0454). At days 14 and 21 p.i., the dissemination rates were ≥97% for all ZIKVs in urban-PP mosquitoes, with no significant differences (Appendix A). The dissemination rates in rural-KC mosquitoes were homogenously high at all-time points and for all ZIKVs (≥ 93%; Figure 2D; Appendix A).

#### 3.2.3. ZIKV Transmission Rate

The transmission rates, determined as mosquitoes with ZIKV-positive saliva among mosquitoes with disseminated infection, were higher for the African ZIKV HD78788 compared with the two Asian isolates at all-time points (urban-PP: Figure 2E, Appendix A; rural-KC: Figure 2F, Appendix A). ZIKV was detected in the saliva of urban-PP mosquitoes as early as day 7 p.i. for HD78788 (6.7%) and NC-2014-5132 (4.2%) strains, and from day 10 p.i. for FSS13025 (6.3%). The transmission rate for HD78788 increased over the following time points (day 10 p.i.: 50.0%; day 14 p.i.: 73.3%) to 100% at day 21 p.i., and was significantly higher than NC-2014-5132 (day 10 p.i.: 4.2%, *p* = 0.0002; day 14 p.i.: 17.2%, *p* < 0.0001; day 21 p.i.: 33.3%, *p* = 0.0019) and FSS13025 (day 10 p.i.: 6.3%, *p* = 0.0033; day 14 p.i.: 10.0%, *p* < 0.0001; day 21 p.i.: 40.9%, *p* = 0.0084). For rural-KC mosquitoes, ZIKV was detected at higher rates, with transmission rates significantly higher for HD78788 on day 7 p.i. (36.7%), day 10 p.i. (80.0%), and day 14 p.i. (96.6%) than for NC-2014-5132 (day 7 p.i.:3.7%, *p* = 0.0028; day 10 p.i.: 18.5%, *p* < 0.0001; day 14 p.i.: 33.3%, *p* < 0.0001). The transmission rate of HD78788 was also significantly higher when compared with FSS13025 on day 10 p.i. (36.7%, *p* = 0.0014). The transmission rate of FSS13025 increased notably over different time points and was significantly higher than the rate for NC-2014-5132 at day 14 p.i. (*p* = 0.0003).

#### 3.2.4. ZIKV Transmission Efficiency

The ZIKV transmission efficiency represents the number of mosquitoes with ZIKV-positive saliva among the total number of tested mosquitoes. The transmission efficiency rates were similar to the transmission rates. However, a comparison between the two mosquito populations revealed significantly higher transmission efficiencies for the rural-KC mosquitoes (Table 1).

### 3.3. ZIKV Viral Load in Infected Mosquitoes

We also investigated the viral load in the artificially infected mosquitoes. We observed an increase in the viral load in the saliva over time in both populations (urban-PP: Figure 3A; rural-KC: Figure 3B). We observed the same pattern for the dissemination into the legs, wings, and head (Appendix A), whereas the viral load in the bodies of the mosquitoes stayed constant (Appendix A). However, the viral load in the saliva, as well as the legs and wings, was higher for the rural-KC mosquitoes. As seen for the transmission rates and the transmission efficiency, the viral load in the saliva was significantly higher for ZIKV HD78788. In urban-PP mosquitoes, we observed significantly higher viral loads for HD78788 (Figure 3A) on day 10 p.i. (*p* < 0.0001) and day 14 p.i. (*p* < 0.0001) compared with both Asian ZIKVs, and on day 21 p.i. compared with FSS13025 (*p* = 0.0007). In rural-KC mosquitoes, the viral loads of the saliva (Figure 3B) were significantly higher for HD78788 than for NC-2014-5132 on day 7 p.i. (*p* = 0.0021), day 10 p.i. (*p* < 0.0001), and day 14 p.i. (*p* < 0.0001), and higher than for FSS13025 on day 14 p.i. (*p* < 0.0001) and day 21 p.i. (*p* = 0.0034). Furthermore, the viral loads in the saliva of FSS13025-infected mosquitoes were significantly higher than for NC-2014-5132 on day 10 p.i. (*p* = 0.036) and day 14 p.i. (*p* = 0.0105). Finally, we observed a positive correlation between the transmission efficiency and viral load in the saliva (urban-PP: Figure 3C; rural-KC: Figure 3D; r^2^ > 0.5 and *p* < 0.0001; Spearman correlation).

### 3.4. Survival Rate of ZIKV-Infected Mosquitoes

During the vector competence study, we observed unusually high numbers of deaths for mosquitoes infected with HD78788, with a survival rate of 6.4% for urban-PP and 11.2% for rural-KC mosquitoes until day 21 p.i. (Appendix A). These survival rates were significantly lower than for urban-PP mosquitoes infected with NC-2014-5132 (18.1%, *p* = 0.003) or FSS13025 (14.5%, *p* = 0.0326), and rural-KC mosquitoes infected with FSS13025 (28.4%, *p* = 0.0005). Overall, the rate of survival was the highest in rural-KC mosquitoes infected with FSS13025 (28.4% after 21 days p.i.). This was significantly higher compared with rural-KC mosquitoes infected with NC-2014-5132 (13.1%, *p* = 0.0038) and higher than the survival of urban-PP mosquitoes infected with FSS13025 (14.5%, *p* = 0.0076). As these findings were unexpected, a comparison of the uninfected mosquitoes are not possible as we did not include a matched number of uninfected mosquitoes over the duration of the experiments.

### 3.5. ZIKV Replication in Neonatal Swiss Mice

The comparative analysis of the pathogenesis in mammals for the different ZIKV strains was done by infecting neonatal Swiss mice intracerebral and monitoring their weight (Figure 4A), general health condition (Appendix A), and survival rate (Figure 4B) over 21 days or until death. All of the mice infected with HD78788 (*n* = 20) died before the end of the experiment (between day 6 and 10 p.i.). Until day 6 p.i., these mice increased their weight similar to the mice infected with the Asian ZIKVs and the control. However, from day 6 p.i. onwards, the health of mice infected with African ZIKVs deteriorated until 10 days p.i. In contrast, the survival rates of mice infected with the Asian lineage strains were much higher, with mortality rates for NC-2014-5132 and FSS13025 infection of 6% and 15%, respectively, at the end of experiment. In addition, the health of the mice was less affected as they only appeared weaker than the controls when infected with NC-2014-5132, or showed limb weakness when infected with FSS13025. The mice infected with FSS13025 exhibited more weight loss from day 14 p.i. onwards compared with the mice infected with NC-2014-5132 and the control mice.

The viral load in the blood and several organs was analyzed by RT-qPCR after death or euthanasia (Appendix A). All of the ZIKVs were replicated in the blood and all of the tested organs, with the highest viral loads in the brain followed by the blood. The viral loads in blood and most organs were higher at day 7 p.i. and were significantly higher at day 8 and 10 p.i. for strain HD78788 than for NC-2014-5132 and FSS13025 (after euthanasia at day 7 and 8-10 p.i.). For HD78788, the viral load was the highest in the brain (median: 648, 3730, 1080 FFU/mL, 643-653, 713-7720, 361-16900 FFU/mL 95% CI at day 7, 8-10 p.i, respectively) and the blood (median: 10, 1575, 642 FFU/mL, 6-14, 3-7910, 5-1280 FFU/mL 95% CI at day 7, 8-10 p.i, respectively; Appendix A).

## 4. Discussion

We investigated the replication of different ZIKV lineages in vitro and in vivo and observed significant differences between the African lineage strain HD78788 and the Asian lineage strains NC-2014-5132 and FSS13025. The discrimination into the two lineages is based on the phylogenetic analysis and geographic origin [28]. The rapid spread and clinical presentation with severe neurological and congenital abnormalities raised the question of whether the Asian lineages are phenotypically different from the African viruses. Our growth analysis in simian and mosquito cells showed that the African virus (human isolate HD78788) was highly pathogenic and induced cell death in all cell lines. This was already reported [28,29,30,31,32,33,34] mainly with the African prototype virus MR766 isolated from the sentinel monkey in 1947 in Uganda [35]. However, this strain has a complex and heterogeneous passage history, and exists in at least three different variations [28] (Genbank NC012532, AY632535, HQ234498). Contrary to our in vitro results, the analysis in human endothelial cells found a faster replication and induction of CPE for the Asian lineage strains (PRVABC59 and FLR) when compared with African strains (MR766 and IbH30656) [36]. Similar findings were reported from human cortical progenitor cells and in human brain organoids infected with a recent ZIKV isolate from Brazil (ZIKV^BR^) compared with ZIKV MR766 [37].

In addition to the analysis of the growth in vitro, we compared the African and Asian viruses for their vector competence in two different populations of Cambodian *Ae. aegypti* mosquitoes (urban and rural). This is of special interest, as the reported circulation of ZIKV in Asia and especially in Cambodia is low [4,16], which is partially explained by the low rate of symptomatic infections [38], the challenging diagnostics in a DENV endemic country [39], and the lack of a ZIKV-specific surveillance program. Therefore, vector competence of Cambodian mosquitoes for different ZIKV strains is highly valuable for risk assessment. Both Cambodian *Ae. aegypti* populations are competent ZIKV vectors with the virus present in saliva as early as 7 days p.i. Our results confirmed multiple former reports of *Ae. aegypti* in other countries and continents as highly competent to transmit ZIKV [10]. Our comparison of different *Ae. aegypti* populations revealed that transmission was observed earlier and in higher rates in the Kampong Cham mosquitoes originated from a rural habitat compared with the urban mosquito population from Phnom Penh. Significant differences in the ZIKV transmission were similarly found for *Ae aegypti* populations from Brazil, USA, and the Dominican Republic, where only the latter were able to transmit all of the investigated ZIKV strains [12]. In that same study, the African strain (DakAR41525) was able to infect, disseminate, and transmit in all of the investigated *Ae. aegypti* populations, whereas the Asian strains (FSS13025 and MEX1-7) showed significantly lower infection and dissemination rates in the Brazil and US mosquito populations. This is similar to our findings, where the African strain showed much higher dissemination rates. Furthermore, our results are in concordance with high ZIKV titers and transmission efficiency observed in *Ae. aegypti* saliva from New Caledonia infected with HD78788 compared with NC-2014-5132 [40]. However, we did not observe major differences in the infection and dissemination rates between the ZIKV lineages and *Ae. aegypti* populations, as we saw the most dissimilarities in the transmission. Differences in dissemination and transmission rates indicate that the different ZIKV strains might differ in their ability to cross the midgut and salivary gland infection and escape barriers. The ZIKV strains might also vary in their replication capacity in the mosquito midgut and other organs [41]. The differences between the ZIKV strains regarding their transmission can also be caused by the innate immunity of mosquitoes [42,43]. Additionally, intrinsic features of mosquito populations such as genetics may implicate in the interactions between mosquito and arboviruses, as was observed in a study in Thailand [44]. The authors observed that wild type *Ae. aegypti* vary in their infectious dose-response to oral DENV. Further investigation on the genetic susceptibility of the two mosquito populations is needed to decipher the observed difference in urban and rural mosquitoes in Cambodia.

Specific virus–mosquito interactions might play a role in the variation of survival rates we observed, depending on the mosquito population and the virus used for infection. We observed a higher pathogenicity in mosquitoes infected with the African strain HD78788. In addition, the urban mosquitoes had lower survival rates across all ZIKV strains compared with the rural mosquitoes. This indicates a co-evolution of the virus and vector in the rural environment, and that ZIKV might have circulated in Cambodia much earlier than 2007, as suggested by retrospective investigations [16], especially as phylogenetic analyses indicate the introduction of ZIKV to Southeast Asia as early as the 1950 s [8].

We further studied the pathogenesis of the different ZIKV lineages in the mouse model infecting neonatal Swiss mice via intracerebral injection. As seen before for the in vitro experiments, as well as for the mosquito vector competence, the African strain HD78788 showed a higher pathogenicity causing 100% lethality within 10 days p.i. This finding is consistent with the results described by Fernandes et al. [45] who also performed this experiment on neonatal Swiss mice where they observed neurological symptoms within 6 days p.i. However, they used ZIKV strain SPH2015, which was isolated in Brazil in 2015. In another study, in 8-week-old interferon-deficient A129 mice infected with ZIKV strain H/PF/2013 (Asian lineage, isolated in 2013 in French Polynesia), a rapid onset of severe clinical signs and death within 7 days p.i. was observed [46]. These findings are similar to our results for the HD78788-infected mice. Moreover, in our study, the Asian lineage strains replicated to lower titers than the African strain and caused fewer clinical symptoms and lower mortality rates. We detected ZIKV viremia in the blood and brain with higher titers than in the other organs, which was also seen in the studies mentioned above. Again, the African strain HD78788 induced higher viral loads in mouse organs than the Asian strains, which reflected our results in cells and in mosquitoes. High replication, especially from African lineage ZIKV strains, was formerly reported from neuronal cell lines as well [29,31,33,47]. The generally less pathogenic phenotype of the Asian lineage viruses might lead to a less acute, but longer persisting infection. Persistence would allow the virus to infiltrate neuronal tissue and cause birth defects, whereas more lethal ZIKV strains (like the African lineage) might cause more miscarriages/stillbirths [48]. Although the Swiss mouse model is sufficient to investigate virus-depend pathogenicity and disease outcomes, other knock-out mice and immunocompetent humanized mouse models that allow ZIKV infection in adult mice through the natural peripheral route, e.g., the hSTAT2 [49] can be utilized to more accurately mimic natural Zika infection and for the development of therapeutics and vaccines.

We had no opportunity to include an American sub-lineage isolate in our comparative analysis. There might be substantial differences between the most recent outbreak viruses in the Americas and the isolates obtained from the epidemics on the Pacific islands or even older Asian isolates [50]. This was indicated in the vector competence of *Ae. aegypti* from Singapore, as the American isolate BEH815744 replicated faster and more successfully compared with the Asian ZIKV H/PF13 [51]. On the other hand, it is questionable whether the introduction of American ZIKV might lead to a wide distribution, as Asian ZIKVs are already circulating and a certain level of immunity exists in the population in Southeast Asia, even if its extent is hard to estimate as seroprevalence studies are missing.

## 5. Conclusions

Overall, our results confirm the different phenotypes of African and Asian ZIKV isolates in different in vitro and in vivo models. We showed that Cambodian *Ae. aegypti* mosquitoes are competent to transmit different ZIKV strains in particular rural mosquitoes, which is of interest for public health, as these mosquitoes are distributed widely across the country. However, the low transmission efficiency for Asian ZIKV strains, especially in the urban *Ae. Aegypti,* might be one factor contributing to the low circulation of ZIKV in Cambodia. Future studies should include American ZIKV strains to test if the introduction of this virus clade could lead to efficient transmission. Additionally, other possible vector species like *Ae. albopictus* should be tested for their vector competence, as this species is also widely distributed in Cambodia. Furthermore, the immunological status of the Cambodian population for ZIKV should be investigated as well as how the immunity for DENV might interfere with ZIKV infection. Our findings highlight the need for intensified, ZIKV-specific surveillance among the human population and vector species.

## Figures and Tables

**Figure 1 microorganisms-09-01250-f001:**
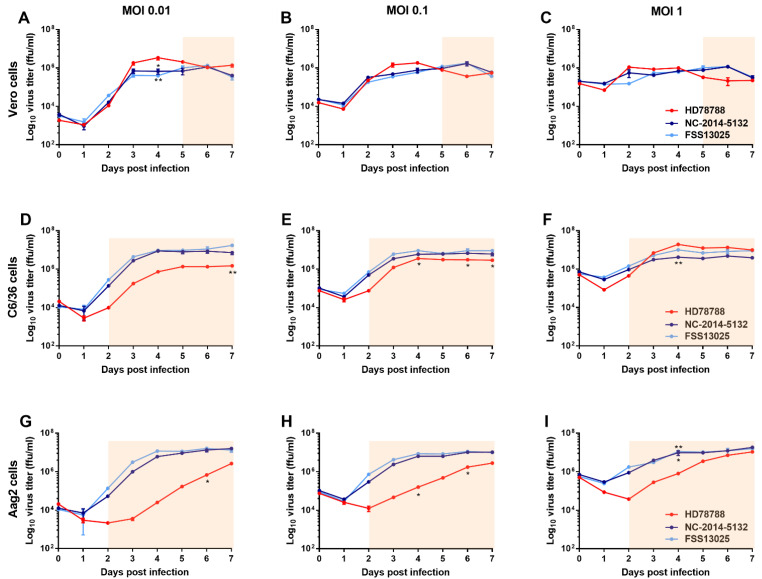
Zika virus (ZIKV) growth kinetic in different cell lines. ZIKV HD78788 (red), NC-2014-5132 (dark blue), and FSS13025 (light blue) were analyzed for their growth in vero (**A**–**C**) and mosquito cells (C6/36: (**D**–**F**); Aag2: (**G**–**I**)). Cells were infected with different multiplicity of infection (MOI): 0.01 (**A**,**D**,**G**), 0.1 (**B**,**E**,**H**), and 1 (**C**,**F**,**I**). Values represent means ± standard deviation of duplicate experiments. Cytopathic effect was observed in cells infected with HD787888 (light red background squares). Statistical differences determined with two-way ANOVA analysis (Tukey’s multiple comparison correction) are marked by asterisks: * *p* < 0.05 and ** *p* < 0.01.

**Figure 2 microorganisms-09-01250-f002:**
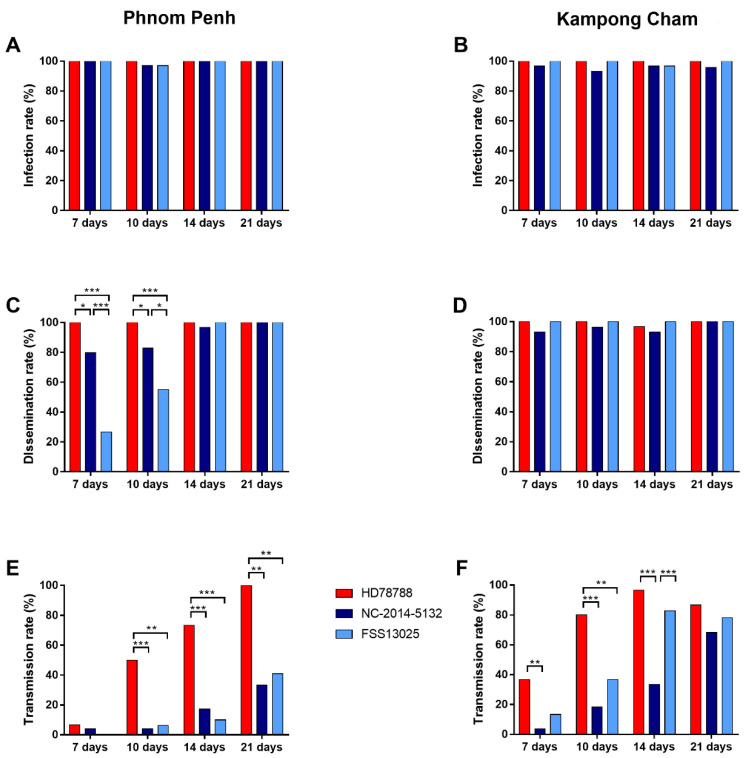
Infection, dissemination, and transmission rates of Cambodian *Ae. aegypti* mosquitoes. *Ae. aegypti* populations from Phnom Penh (**A**,**C**,**E**) and Kampong Cham (**B**,**D**,**F**) were infected with ZIKV HD78788 (red), NC-2014-5132 (dark blue), or FSS13025 (light blue). Infection rates (**A**,**B**) are expressed as the percentage of mosquitoes with ZIKV-positive bodies among the total number of blood-fed mosquitoes. Dissemination rates (**C**,**D**) are expressed as the percentage of mosquitoes with ZIKV-positive head, legs, and wings among the mosquitoes with ZIKV-positive bodies. Transmission rates (**E**,**F**) are expressed as percentage of mosquitoes with ZIKV-positive saliva among mosquitoes with ZIKV-positive bodies. Statistical differences determined with Fisher’s exact test are marked by asterisks: * *p* < 0.05, ** *p* < 0.01, and *** *p* < 0.001.

**Figure 3 microorganisms-09-01250-f003:**
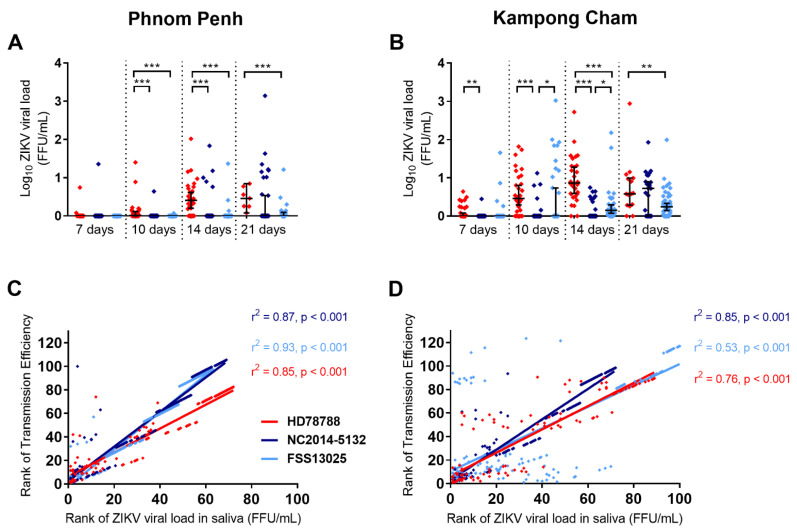
Viral load in the saliva of *Ae. aegypti* mosquitoes. *Ae. aegypti* populations from Phnom Penh (**A**,**C**) and Kampong Cham (**B**,**D**) were infected with ZIKV HD78788 (red), NC-2014-5132 (dark blue), or FSS13025 (light blue). The viral loads in saliva (**A**,**B**) were determined by RT-qPCR and the individual results are plotted with a median and 95% CI. Statistical differences determined with Kruskal–Wallis test (Dunn’s multiple comparison correction) are marked by asterisks: * *p* < 0.05, ** *p* < 0.01, and *** *p* < 0.001. Correlation between the transmission efficiency and viral loads in saliva were evaluated by the Spearman correlation analysis (**C**,**D**).

**Figure 4 microorganisms-09-01250-f004:**
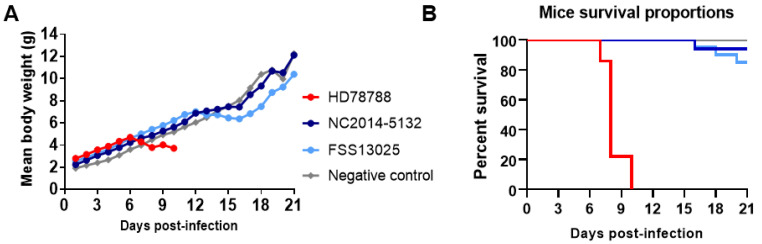
ZIKV infection in neonatal Swiss mice. Mice were infected with ZIKV HD78788 (red), NC-2014-5132 (dark blue), or FSS13025 (light blue) at the age of one day old. Body weight (**A**) was checked every day and survival was monitored for 21 days (**B**).

**Table 1 microorganisms-09-01250-t001:** ZIKV transmission efficiency of two Cambodian *Ae. aegypti* populations.

Days after Infection	Mosquito Population (Origin)	HD78788		NC-2014-5132		FSS13025	
		Transmission Efficiency *	*p* Value ^#^	Transmission Efficiency *	*p* Value ^#^	Transmission Efficiency *	*p* Value ^#^
7	urban-PP	6.7% (2/30)	0.0102	3.3% (1/30)	1.000	0% (0/30)	0.1124
	rural-KC	36.7% (11/30)		3.3% (1/30)		13.3% (4/30)	
10	urban-PP	50.0% (15/30)	0.0292	3.3% (1/30)	0.1945	3.3% (1/30)	0.0025
	rural-KC	80.0% (24/30)		16.7% (5/30)		36.7% (11/30)	
14	urban-PP	73.3% (22/30)	0.0797	16.7% (5/30)	0.3604	10.0% (3/30)	<0.0001
	rural-KC	93.3% (28/30)		30.0% (9/30)		80.0% (24/30)	
21	urban-PP	100% (7/7)	1.000	33.3% (10/30)	0.0283	40.9% (9/22)	0.0031
	rural-KC	86.7% (13/15)		68.2% (15/23)		78.0% (39/50)	

* Number of mosquitoes with ZIKV positive saliva/total number of blood-fed mosquitoes. ^#^ Two-tailed Fisher’s exact test; urban-PP—urban Phnom Penh; rural-KC—rural Kampong. Cham.

## Data Availability

Publicly available datasets were analyzed in this study. This data can be found here: https://drive.google.com/drive/folders/1-0OySA75-bw9nMq2uDhRH7m-9YrJ7v7q.

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
