# Peer review of "Replication Variance of African and Asian Lineage Zika Virus Strains in Different Cell Lines, Mosquitoes and Mice"

_microorganisms, 2021, doi:10.3390/microorganisms9061250_

Round 1
Reviewer 1 Report
The replication in Vero, C6/36 and Aag2 cells were compared in three zika virus strains. The viral growth in Vero cells were similar among the three viruses but the growth of HD78788 was lower than other two viruses in C6/36 and Aag2 cells. The CPE was only observed after HD78788 infection in C6/36 and Aag2 cells. Authors explained the strong cell damage inhibited the viral growth after HD78788. The rapid or high viral replication at early time point was not the reason for the severe CPE after HD78788 infection with the similar or lower viral titers at day0-2 after HD78788 infection (Fig 1D-I). What was the reason for severe CPE after HD78788 infection in C6/36 and Aag2 cells? When did the CPE appear then? Only African zika virus caused CPE in lines 199 and 212-213 in the manuscript, while in the previous report (PLoS One. 2017 Sep 12;12(9):e0184397.) Asian zika virus infected C6/36 cells showed cytolysis, cellular individualization, and detached cells in the supernatants. How the authors explain the difference? Authors need to provide the source of mosquito cells in Materials and Methods. HD78788 replicated less in mosquito cell lines whereas the viral titers in bodies and wings were higher in HD78788 infected mosquitos compared to Asian viruses (Fig S2). How authors explain this discrepancy? What was the biological difference rural and urban mosquitos? Why all three viruses caused higher transmission rate in rural mosquitos? The comparison between African and Asian zika viruses is well studied previously as authors described. Authors should more focus on the differences the vector competence of rural and urban mosquitos. The authors should provide the reference for lines 36-38. The authors should discuss why there was the contradiction in lines 360-365. The lines 422-424 was not supported from the present study.
Reviewer 2 Report
Tey Putita Ou et al. wrote an interesting article about different strains of ZIKV replication on mosquitoes and mice. Even though it looks like a lengthy article but written very interestingly and it is important to understand the vector competence for different strains of ZIKV at least for geographical regions because that decides the infectivity.
Reviewer has the following minor comments:
- Is there any reason to believe that the African strain grown in AP-61 cells made the virus more virulent than the other two viruses that grew in C6-36 cells?
- Maybe in figure.1, on top of each column, authors can specify the MOI, instead of reading on the legends. That will help the reader to better understand the graphs.
- Reviewer wonder than is that 7 days to see the virus in mosquitoes with ZIKV-positive body, or has it been observed earlier than that? Because Figure 2, shows the infection rate, dissemination rate, and transmission rate are all happening at the same time point. Alternatively, is there any reason that the authors collected the samples from 7 dpi onwards only? It would be interesting if the authors had any data from earlier time points that would provide the kinetics of infection and dissemination rate.
Reviewer 3 Report
In this manuscript, Tey Putita Ou and co-authors investigate the replication of African and Asian characterized Zika viruses in vitro and in vivo. The findings are relevant to the field and this reviewer has no observation but a comment: thank you for your research contribution.
Round 2
Reviewer 1 Report
1 There is contradiction in authors’ comments. For first question, “What was the reason for severe CPE after HD78788 infection in C6/36 and Aag2 cells?”, the authors explained the HD78788 adapted to Vero cells and this may have caused its less severe CPE to mosquito cells. However in Fig 1B, C the viral growths of three viruses were similar in Vero cells. 2 For second question, the authors said the increased viral replication mediated the cell death of the mosquito cells. This is not consistent with the data high viral replication of NC-2014-5132 and SS13025 did not cause the cell damage. 3 I do not understand how 1) the different source of organs in vitro and in vivo, 2) lower survival rate of infected mosquitos with HD78788, and 3) unknown passage history, explain the discrepancy in vivo and in vitro. Also authors need to modify the manuscript according to the reviewer’s comments not only answering them in the letter. 4 The authors did not provide any comment to the reviewer’s comment, “The comparison between African and Asian zika viruses is well studied previously as authors described. Authors should more focus on the differences the vector competence of rural and urban mosquitos.” 5 Authors should provide the data or at least references to show what the difference is in rural and urban mosquitos. Otherwise there was no scientific reason to compare the replication differences in these to mosquito populations and the reviewer has to say the research design was not appropriate.Author Response
Please see the attachment.
